# Radio-Guided Lung Surgery: A Feasible Approach for a Cancer Precision Medicine

**DOI:** 10.3390/diagnostics13162628

**Published:** 2023-08-09

**Authors:** Miriam Conte, Maria Silvia De Feo, Viviana Frantellizzi, Miriam Tomaciello, Francesco Marampon, Laura Evangelista, Luca Filippi, Giuseppe De Vincentis

**Affiliations:** 1Department of Radiological Sciences, Oncology and Anatomo Pathology, Sapienza University of Rome, 00161 Rome, Italy; 2Department of Biomedical Sciences, Humanitas University, Via Rita Levi Montalcini 4, 20072 Pieve Emanuele, Italy; 3IRCCS Humanitas Research Hospital, Via Manzoni 56, 20089 Rozzano, Italy; 4Department of Nuclear Medicine, Santa Maria Goretti Hospital, 04100 Latina, Italy

**Keywords:** radio-guided surgery, GGO, lung nodule, lymphadenectomy, ^18^F-FDG, technetium, ^99m^Tc albumin macroaggregates, indium-111

## Abstract

Background: Radio-guided surgery is a reliable approach used for localizing ground-glass opacities, lung nodules, and metastatic lymph nodes. Lung nodules, lymph node metastatic involvement, and ground-glass opacities often represent a challenge for surgical management and clinical work-up. Methods: PubMed research was conducted from January 1997 to June 2023 using the keywords “radioguided surgery and lung cancer”. Results: Different studies were conducted with different tracers: technetium-99m-albumin macroaggregates, cyanoacrylate combined to technetium-99m-sulfur colloid, indium-111-pentetreotide, and fluorine-18-deoxyglucose. A study proposed naphthalocyanine radio-labeled with copper-64. Radio-guided surgery has been demonstrated to be a reliable approach in localizing a lesion, and has a low radiological burden for personnel exposure and low morbidity. The lack of necessity to conduct radio-guided surgery under fluoroscopy or echography makes this radio-guided surgery an easy way of performing precise surgical procedures. Conclusions: Radio-guided surgery is a feasible approach useful for the intraoperative localization of ground-glass opacities, lung nodules, and metastatic lymph nodes. It is a valid alternative to the existing approaches due to its low cost, associated low morbidity, the possibility to perform the procedure after several hours, the low radiation dose applied, and the small amount of time that is required to perform it.

## 1. Introduction

Ground-glass opacities (GGOs) represent a challenge for diagnostic classification and clinical assessment. The major incidence of these phenomena depends on the development of more sensitive diagnostic tools that are currently available. The necessity to exclude the concomitant presence of oncological disease in the context of a GGO remains a complicated task. GGOs can be determined based on non-oncological conditions such as interstitial focal fibrosis, eosinophilic pneumonia, obliterating bronchiolitis, endometriosis, Wegener’s granulomatosis, or aspergillosis [1], and can include different histological types of lung cancer (such as in situ adenocarcinoma; minimally invasive adenocarcinoma; and, less frequently, invasive adenocarcinoma) [2]. Under computed tomography (CT) imaging, a solid component in a GGO indicates the presence of a neoplasm. When the solid component disappears after an antibiotic therapy or steroid treatment, it underlays benign conditions. The dimension’s stability is also associated with localized fibrosis or atypical adenomatous hyperplasia (AAH) [3]. GGOs can be a manifestation of an alveolar bronchial carcinoma (BAC) or an adenocarcinoma. According to some authors, such as the survey from Lee et al., suspicions surrounding the presence of neoplasm should be always taken into account. They classified GGOs as pure nodular GGOs (PNGGOs) when no lung parenchyma was obscured at CT and mixed nodular GGOs (MNGGOs) when patches obscured lung parenchyma. The authors analyzed 55 lesions in 23 patients with PNGGOs of 10 mm, while 13 lesions in 13 patients were PNGGOs > 10 mm. The 11 lesions identified in 8 subjects were MNGGOs 10 mm and the 17 lesions in 16 patients were MNGGOs greater than 10 mm. Histology identified six cases of AAH, five cases of BAC, and one case of focal interstitial fibrosis for PNGGOs around 10 mm. PNGGOs > 10 mm included two cases of AAH, two cases of BAC, two adenocarcinomas, and four cases of interstitial fibrosis. MNGGOs around 10 mm were also classified: two cases of adenocarcinomas and one case of BAC, while MNGGOs > 10 mm included eleven cases of adenocarcinoma, three cases of BAC, and only one case of aspergillosis [4].

For the metabolic characterization of a similar lesion, fluorine-18 fluorodeoxyglucose ([^18^F]F-FDG) photon emission tomography (PET) CT is often required, even if the low specificity in the differential diagnosis between malignant and inflammatory conditions makes its use debatable. As an example, BAC appears as a GGO at CT imaging, but it is associated with the low or absent uptake of [^18^F]F-FDG during PET imaging due to its moderate degree of nuclear atypia and mitotic figures, as well as the presence of desmoplasia and necrosis [5,6]. Other GGO lesions with a low glucose uptake are lung metastasis from breast cancer or renal carcinoma, as well as the mucinous adenocarcinomas of the gastrointestinal tract that are characterized by mucinous components and low cellularity [7].

A relevant pathological entity is the solitary pulmonary node (SPN), defined as single and spherical with well-defined boards and a diameter less than 3 cm, located in lung parenchyma and not combined with atelectasis, adenopathy, or pleural effusion. Over the last few decades, the SPN showed an increase in its detection thanks to the implemented use of CT [8]. The probability of malignancy in SPNs is related to imaging characteristics and the patient’s risk factors [9]. Since the prognosis depends on the histology of the lesion, the differential diagnosis between benign and malignant nodes is fundamental. Albeit a follow-up with a CT scan for small nodules has been proposed, especially when the dimensions are less than 5 mm, an increasing rate of malignancy has been detected in nodules bigger than 5 mm [10]. For smaller lesions, it is important to establish a diagnosis when present in high-risk patients (i.e., oncological populations). Percutaneous biopsy is not often feasible in sub-centimetric lesions [11] and [^18^F]F-FDG has a limited value for sub-centimeter nodules, mucinous adenocarcinoma with low cellularity, and low-grade malignancy [12]. 

The advent of video-assisted thoracoscopic surgery (VATS) and robotic approaches has changed the management of these lesions since they represent the first choice for peripheral lung nodules, even if its application becomes challenging in the case of small, nonsolid, or deep lung nodules [13].

Another important aspect is the resection of lymph nodes since their involvement influenced the prognosis according to TMN classification. The benefits with respect to the identification of sentinel lymph nodes (SLNs) in patients with different solid tumors, such as breast cancer and melanoma (in which the SNL dissection predicts the status of the more distant lymph nodes stations) are well known. On the other hand, the application of this technique is still debated in lung cancer. In fact, international recommendations suggest that a complete lymph node resection has been conducted and no changes have been made so far. Moreover, enlarged lymph nodes are present in 40% of benign cases, while normal nodes can be involved in 25% of cases. Moreover, it is common that some nodal stations can be skipped, so the standard surgical operation practice involves conducting a complete lymphadenectomy that is not burdened with morbidity [14,15,16,17,18,19]. In addition, 40% of early lung cancer cases have micrometastasis at surgery time [20]. Therefore, some studies have suggested radiological techniques for the identification of sentinel lymph nodes in early-stage lung cancer, as discussed below.

Among these pathological conditions, lung neuroendocrine tumors represent a notable opportunity for the application of radio-guided surgery [21].

The intraoperative localization of these entities through the support of radiopharmaceutical agents represents a reliable and easy technique that can be used to identify lung lesions and obtain a satisfactory surgical resection, as well as to enhance post-operative courses [11].

Different radiopharmaceuticals have been used for radio-guided surgery. PubMed research was conducted from January 1997 to June 2023 using the keywords “radioguided surgery and lung cancer”. The studies in question focused on radio-guided surgery for lung nodules, GGOs, or thoracic lymph nodes in the context of lung cancer. This review aims to summarize the application and the most relevant agents developed for the radio-guided approach in lung nodules.

## 2. ^99m^Tc-Albumin Macroaggregates in the Radio-Guided Surgery of Lung Lesions

The application of [^99m^Tc]-albumin macroaggregates (MAAs) represents a cornerstone of radio-guided surgery since it is a widely used agent in lung scintigraphy. The use of macroaggregates seems to reduce the lung parenchymal spreading of radiopharmaceuticals, as observed by Daniel et al. [22] and Galetta and colleagues [11].

The radio-guided occult lesion localization (ROLL) of pulmonary nodules represents an additional option to hook wire. It consists of the injection of the agent into the lesion under CT guidance and an intraoperative localization through a gamma probe. Even if frequently used in breast cancer [23], several studies have applied this method to lung cancer, as reported below. In a study conducted by Vollmer et al. [24], 38 nodules, including 36 lung nodules, 1 nodule of the chest wall, and 1 pleural one, were injected with [^99m^Tc]-MAA (37-111 MBq) 2–18 before surgery. Two different gamma probes were used: a 14 mm diameter hand-held gamma detection probe for superficial nodules and a 10 mm diameter device for the deeper nodules. Only non-palpable solid nodules with a dimension ≥ 5 mm, subsolid nodules with a solid component ≥ 5 mm, or other nodules of the chest (which were difficult to localize through palpation) were considered. All the lesions were efficiently detected and only in two cases, VATS was converted to thoracotomy because of the deep position of the lesion or the laceration of the lung during the procedure. The mean length of the VATS was 19 min. Six benign lesions were identified, while 8 were primary lung cancers and 24 were metastasis.

Carvajal et al. [25] conducted a retrospective study based on data recorded from patients with lung nodules treated with uniportal VATS (UVATS). The included patients had nodule sizes smaller than 1 cm (with a median size of 7 mm), characterized by a subsolid morphology and a distance from the pleura greater than 1 cm. They injected 7–10 MBq of [^99m^Tc]-MAA with 0.1 mL of non-ionic contrast into or near the nodule. In total, 16.6% of patients were affected by malignant neoplasm, 41.7% had lung metastases, and the other 41.7% had benign lesions. The rate of pneumothorax caused by tracer injection was 8.3%. Statistical analysis highlighted how the concordance between palpation and the presence of a nodule was poor. In fact, 91.7% of patients had non-palpable nodules during surgery.

Dailey et al. [26] applied the radiotracer-assisted localization of lung nodules (RALN) with UVATS. Under intermittent CT fluoroscopic guidance, 300 µCi of [^99m^Tc]-MAA was injected into 29 lung lesions. Furthermore, 86.5% of nodules were malignant; in particular, 11 were invasive adenocarcinoma, 2 were squamous cell cancer, and the other 2 were in situ adenocarcinoma. The other 10 lesions were metastatic tumors of sarcoma, colorectal cancer, hepatocellular cancer, and melanoma. The remaining four were benign. The average time of radiopharmaceutical injection to surgery was 219 min. The authors affirmed that RALN associated with UVATS was an effective approach that could be used to resect small and deep pulmonary lesions that could be difficult to identify during thoracoscopy or with palpation. A notable aspect is the radiation exposure of involved personnel. The authors observed that when compared to intraoperative fluoroscopy or CT imaging, radiation exposure to the operating room and laboratory employees was relatively lower with radiotracer-assisted surgery.

Durmo and colleagues [27] developed a fluoroscopy system on a single photon emission computed tomography/CT (SPECT/CT) scan, which was useful in order to guide the injection of the tracer. Moreover, 15 MBq of [^99m^Tc]-MAA was injected into the nodule displayed at chest CT without contrast, and then a needle under CT-guided fluoroscopy was used to evaluate the distance between the lesion and the skin. The vessels then interposed in order to not cross two different lobes. Totally, 37 lung nodules were identified; 15 were GGOs, 12 solid, and 10 were partially solid. Of them, 42% had a histology indicative of primary lung cancer, 22% were metastasis, and 36% were benign. The procedure was reliable, with a high success rate, minimal complications, and no mortality or relevant morbidity. Moreover, [^99m^Tc]-MAA has a non-significant radiation exposure rate for patients and hospital personnel.

The same radiopharmaceutical was chosen by Bertolaccini et al. [28] who, under CT guidance, injected 2–9 MBq of [^99m^Tc]-MAA near or into the lung lesion. The inclusion criteria were a maximum nodule diameter less than 15 mm, a distance from the pleura of 20–40 mm, the presence of a GGO, and/or a posterior location of difficult-to-detect SPNs through palpation. The resection was conducted through VATS and a gamma counter allowed the lesion to be localized and the suture margins to be checked for residual uptake. The authors observed that the radio-guided approach allowed the identification and resection of GGOs, which made it difficult to resect through VATS alone, and, even if a multi-disciplinary collaboration was required, no complexity and expenses were added. Moreover, the method did not influence the execution of a histological exam. As stated before, the low patient-absorbed radiation dose of the technetium, thanks to its 6 h half-life, makes this tracer a safe choice for radio-guided surgery.

In 2000, Boni et al. tested the execution of a radio-guided VATS for peripheral lung nodules using 5–10 MBq of [^99m^Tc]-labeled human serum albumin microspheres [29]. The lesion was identified using a collimated probe (11 mm in diameter), connected to a gamma ray detector, and introduced in the trocar during thoracoscopy. Thirty-nine patients were enrolled with nodules characterized by a mean size of 8.3 mm. The histological exam proved 18 malignant cases and 21 benign lesions. Six cases of pneumothorax were found, but no other complications were reported.

As observed in the previous study, radio-guided surgery remains a feasible and precise technique that can be used to exactly localize and detect the border of a lung lesion, both nodules, and GGOs. It was not affected by the issue of percutaneous hook wire and methylene blue injection. For example, dislocation for the hook wire or the disappearance of interval marks between methylene injection and surgery can take too long, and hemothorax, pneumothorax, and chest pain are more common with the other two alternatives.

The superiority of the radio-guided procedure was also reported in another study by Gonfiotti et al. [30]. They included 50 subjects with SPNs which were clustered into two groups: one half underwent a hook wire procedure (group A) and the other underwent radio-guided surgery (including the injection of 5–10 MBq of microspheres of human albumin serum labeled with [^99m^Tc]) (group B). The minimal used activity did not require any special exposure reading for the staff. The surgical procedure was performed 120 min to 16 h after the tracer injection. The radio-guided localization was more effective than the hook wire approach (96% vs. 84% in the first and second group, respectively) and finger palpation (28% and 24% in the first and second group, respectively). The pneumothorax was more frequent in group A, thus demonstrating that radio-guided surgery was associated with fewer complications. 

Galetta et al. [11] enrolled 112 patients with subsolid nodules smaller than 1 cm (with a mean size of 9 mm) and with a distance from the visceral pleura of 1 cm or both. Based on surgery time, 7 to 15 MBq of [^99m^Tc]-MAA was injected 24 h before the surgical intervention. In total, 88.6% were malignant lesions, while 11.4% were benign formations. 

Different methods have been used to address the challenges of localizing smaller, deeper, and GGO nodules. The study conducted by Manca et al. documents 20 years of experience with radio-guided thoracoscopic surgery [31].

Under CT guide guidance, 5 MBq of [^99m^Tc]-MAA for the 1-day procedure to 15 MBq of [^99m^Tc]-MAA for the 2-day procedure was injected in 395 patients with SPNs smaller than 2 cm (with a mean size of 13 mm) and deeper than 5 mm below the visceral pleura (in a range of 6–29 mm). Excluding 12 patients who reported asymptomatic pneumothorax, no major complications were reported. During surgery, the area with the highest uptake was localized through an 11-milimeter-diameter collimated thoracoscopic gamma probe. The operation lasted 40 min on average with a mean time of 3 min for SPNs localization with the gamma probe. Histological examination confirmed that the 395 nodules were made up of the following: 206 benign lesions, 130 metastasis cases, and 59 lung primary tumors. The lack of close timing-related problems centered around the preoperative identification of nodules made the radio-guided procedure a better option compared to the hook wire approach and dye injection. The procedure could be performed up to 24 to 36 h before surgery and the staff did not need a special radiation exposure reading since the used radiopharmaceutical agent’s dose was minimal.

This was previously stated in a study conducted by Zaman and colleagues [32], in which the superiority of radio-guided thoracoscopic surgery was demonstrated when it was compared to hook wire, dye, and endothoracic ultrasonography.

Nineteen patients with non-palpable SPNs (with a diameter smaller than 15 mm and a distance from the nearest pleural surface of 20–40 mm) were included in Bertolaccini et al.’s study [33]. Also, a posterior location and a non-solid or partly solid morphology were considered. In addition, 2–9 MBq of [^99m^Tc]-MAA injection fluid, guided by CT imaging and surgery, was executed 3 to 19 h after the nodule’s tracer was localized. All the nodules were identified through a gamma probe during surgery with a mean time of 6 min. Eight cases showed primary lung malignancies, four cases showed secondary lesions, and the remaining seven cases showed a benign nodule.

The large size of the population was featured in the work of Ricciardi et al. [34]. They included 451 patients with GGOs or SPNs (with a diameter less than 20 mm) and a distance from visceral pleura superior or equal to 5 mm and within 3 cm. Furthermore, 5–10 MBq of [^99m^Tc]-human serum albumin microspheres was injected into the lesion with a non-ionic contrast agent to ensure the possibility of guiding the radiopharmaceutical injection through a CT scan. The surgery was performed with a mini-invasive resection VATS or robotic-assisted thoracoscopic surgery (RATS) through da Vinci’s third- and fourth-generation systems. VATS/RATS were realized 60 min to 20 h after the radiological procedure, with a mean duration time of 40 min and a lesion radio localization time of 3 min. The success rate of the procedure was 98%. No mortality was recorded at 30, 60, or 90 days after surgery. Histological examination showed that 48.34% had benign lesions, 33.04% had metastasis, and 18.62% had primitive lung cancer. Moreover, the cost of radio-guided surgery was limited to EUR 30 (USD 32.95) per patient, which is lower than other approaches. Finally, as observed in previous studies, the low dose of the tracer made this procedure safe for the involved personnel.

In the 2018 paper by Davini and colleagues [8], 175 enrolled patients were found to be affected by SPNs with a diameter less than 10 mm and/or a distance from the nearest pleural surface major or equal to 10 mm. A CT-guided [^99m^Tc]-human serum albumin microspheres injection was performed. VATS resection was conducted within 3 h from the tracer injection. Only 13 patients suffered from pneumothorax after the procedure without needing a chest tube insertion. The radio-guided surgery permitted a 100% localization of nodules and an average time length of 44 min. Fifteen lesions were benign while 120 were malignant. In this study, the choice of ^99m^Tc (with its low absorbed radiation) made this procedure safe for the patient and for the involved staff.

In Table 1, the up-cited studies were summarized.

## 3. Radio-Guided Sentinel Lymph Node Biopsy

In the early stage of non-small cell lung cancer (NSCLC), the detection of SLNs has been proposed to reduce complete lymphadenectomy. Melfi et al. [14] enrolled 29 patients with stage IA-IB NSCLC. A dose of 37 MBq of [^99m^Tc]-nanocolloid was injected near the tumor during surgery in 10 patients under CT guidance in the remaining patients. The study showed that the optimal time to conduct surgery was 1 h after tracer injection. The SLNs were localized in 25/26 cases, with 31 SLNs identified in total, 7 of which were metastatic. The approach resulted in a feasible approach used to identify SLNs in NSCLC. 

In a study by Rzyman and colleagues [35], 110 patients with clinical N0 NSCLC underwent radio-guided surgery using four random types of tracers: [^99m^Tc]-human albumin (particle size: <80 nm), sulfur colloid (particle size: 32 to 178 nm), filtered sulfur colloid (particle size: <50 nm), and tin colloid (particle size: 100 to 3000 nm).

The tracer was injected in the four quadrants of the peritumoral area. The radio-guided approach showed a detection rate of 100%, a sensitivity rate of 87%, and a negative predictive value of 93%. Any significant differences were observed among tracers, even if smaller particles performed better.

## 4. Cyanoacrylate Combined with Technetium-99m-Sulfur Colloid (Sn-^99m^Tc)

Cyanoacrylate, in association with [^99m^Tc] sulfur colloid (Sn-99m Tc) en dash sign, was injected by Tyng et al. in a patient with a GGO for the first time in 2015 [36]. The mean activity of the agent ranged between 18.5 and 37 MBq. This approach’s innovation rests on using cyanoacrylate which polymerizes, immediately forming a nodule. This inhibits the radiotracer dispersion into the lung parenchyma [37] and, when a tracer is added, permits the identification of the nodule and its margin with a gamma probe (either during open lung biopsy or during VATS). The authors applied this method to two cases of GGO (the first with dimensions of 18 × 16 mm with a solid component inside of 7 mm localized in the posterior segment of the right upper lobe, while the second was 24 × 22 mm and was situated in the upper segment of the right lower lobe). In the first case, the lesion was promptly identified with a gamma probe, and a segmentectomy with lymphadenectomy was performed. The histological exam gave evidence of a moderately differentiated lepidic-predominant invasive lung adenocarcinoma, with areas of solid and acinar growth patterns in the absence of lymphatic metastasis. In the second case, dextran-^99m^Tc was injected to reveal the sentinel lymph node. Histology showed a non-mucinous in situ adenocarcinoma of the lung, without lymphatic involvement.

As observed by the authors, this approach overcomes the risk of marker dislocation and does not hinder the frozen section and histological examination. It is also characterized by minimal morbidity, with a lower incidence of complications (pneumothorax and hemorrhage) compared to the hook wire technique [30,38,39,40,41].

## 5. Indium-111 (^111^In)-Pentetreotide for Neuroendocrine Tumors

An attractive practice involves using [^111^In]-pentetreotide for detection purposes during neuroendocrine tumor surgery [42]. Grossrubatscher and associates [21] reported a case of a 28-year-old female affected by Cushing’s syndrome due to an ACTH-secreting bronchial carcinoid. After the first surgical procedure to remove the bronchial carcinoid, the disease persisted so pentetreotide scintigraphy was used and showed a mediastinal uptake in the paramedian region. After a thymectomy, scintigraphy was re-performed and revealed a focal uptake in the paramedian region of the mediastinum. Given the ineffectiveness of previous surgery, a radio-guided approach was conducted. A total dose of 148 MBq of [^111^In-DTPA-D-Phe1]-pentetreotide was administered to the patient and she underwent surgery 48 h later. All the remaining thymic tissue and residual mediastinal lymph nodes were removed through the guidance of the gamma probe. After resection, the surgical site was scanned to identify residual activity. No other pathological tissue was detected in the following radiological examination and the patient reached a complete remission. 

Radio-guided surgery with [^111^In]-pentetreotide was used in the case report of Mansi et al. [43]. A case of a 28-year-old patient with Cushing’s syndrome was described. An atypical uptake in the lower lobe of the right lung was seen following pentetreotide scintigraphy. Radio-guided surgery was performed 4 days after a second intravenous administration of 111 MBq of [^111^In]-pentetreotide, and the resection of a small mass (with dimensions of 1.8 × 1.4 × 1.6 cm) was conducted. Histology allowed a lung carcinoid to be diagnosed. The high tumor-to-normal tissue ratio and tumor-to-hilar normal lymph node ratio permitted a complete resection of the tumor, saving non-involved tissue. This result was permitted by the absence of the blood pool activity of the tracer, though was not possible if radio-labeled antibodies were used.

Loli and colleagues [44] reported six occult bronchial carcinoid cases. In this experience, radio-guided surgery with indium–pentetreotide permitted the excision of all the neoplasm, but in one case of remnant disease after two operations, it guided the surgery in the complete excision of residual mediastinal lymphatic disease.

## 6. Fluorine-18 (^18^F)-Deoxyglucose

A handheld high-energy gamma probe (PET-Probe) was proposed for the radio-guided surgery of [^18^F]F-FDG-avid tumors and was able to detect lung metastasis in two patients with melanoma [45,46].

Ten patients with resectable lung cancer were enrolled in a study conducted by Nwogu and collaborators [47]. A total dose of 10 mCi was injected into the patients and radio-guided surgery was conducted within 4 h. All patients underwent preliminary [^18^F]F-FDG PET imaging. A handled gamma detector was used to detect the presence of increased [^18^F]F-FDG uptake in lymphatic stations of the thorax. The device was able to localize all the lymph nodes that were positive at the PET scan. In three patients, the probe detected nodes that were negative to conventional pathological examinations. Four subjects had positive hematoxylin–eosin lymph nodes, but the PET probe localized adjacent nodes with micrometastasis. Only three nodes were false-positive, as demonstrated by the PET probe. However, the challenge to distinguish FDG activity in primary lesions from the signal of the nearest lymph nodes remains; indeed, this pitfall was associated with false-negative N1 nodes in two cases.

## 7. Naphthalocyanine and Cupper-64 (^64^Cu)

A biocompatible marker based on sucrose acetate isobutyrate, ethanol, and a multifunctional naphthalocyanine (NC) dye was developed by Wang et al. [48]. This compound is characterized by near-infrared fluorescence which makes an image-guided resection possible at long tissue depths. Naphthalocyanine dye was also used as a chelator for the ^64^Cu isotope. Radiolabeling with copper did not interfere with the fluorescence of NC. The compound was tested on mice that demonstrated high photostability over 4 weeks and retention for 48 h. The tracer was suggested for soft tissue radio-guided surgery and for lung malignancies with radio-guided operations.

## 8. Discussion

Lung nodules, lymph node metastatic involvement, and GGOs represent a challenge for surgical management. Different methods that better localize these lung lesions were proposed.

Hook wire placement, often in combination with methylene blue injection under CT guidance, was used. However, the dislocation of hook wire and the difficulty in identifying methylene blue marks on the pleural surface represent the main causes of conversion to open surgery [29]. The disappearance of dyeing (when the interval between methylene injection and surgery is too long), hemothorax, pneumothorax, and chest pain are more frequent with the abovementioned approaches. Moreover, the use of non-water-soluble marks seems to lead to an increased risk of embolism and cerebrovascular accidents if they unintentionally reach the systemic circulation [49].

Another feasible technique is ultrasonography (US) during thoracoscopy. However, it is dependent on the experience of the operator and is limited by the presence of air in the lungs, which is responsible for reverberating artifacts. This pitfall could be overcome by filling the chest cavity with saline solution [29]. The US approach is appealing because it does not require coordination with the radiology department. This is a relevant aspect that represents an advantage compared to radio-guided surgery (which needs the cooperation and expertise of a nuclear medicine unit). On the other hand, ultrasound remains a time-consuming approach in pulmonary emphysema patients whose lungs struggle to fully collapse [11].

The feasibility of radio-guided surgery can precisely detect a lesion in the low radiological burden of personnel exposure [50]. In fact, if it is compared with the use of cyanoacrylate, there are different positive aspects. Cyanoacrylate needs fluoroscopy which has various risks for the patients and the staff present in the operating room [37]. Similar considerations could be made for near-infrared-guided indocyanine green (ICG) fluorescence imaging. It has been used in abdominal surgery but is successfully applied for lung nodule localization, intersegmental plane visualization, positive and reverse staining, and target pulmonary artery analysis [51]. It has low intraoperative complications, few side effects, and long persistence in lung parenchyma (even if it remains, this method is not always available in all centers) [51,52].

On the other hand, as observed before, for technetium, which has a low absorbed dose and low costs, radio-guided surgery, especially when realized with technetium, is a safe approach and a valid economic alternative. Moreover, a low morbidity associated with radio-guided surgery has been reported in the previously mentioned studies. The lack of a necessity to conduct radio-guided surgery under fluoroscopy or echography, which needs great experience [11], means this technique makes it easy to perform a precise surgical procedure.

## 9. Conclusions

Radio-guided surgery represents a reliable approach for the localization of GGOs, lung nodules, and metastatic lymph nodes. It is a valid alternative to existing approaches (such as fluoroscopy, intraoperative echography, and hook wire procedures) that are used due to its low cost, low morbidity, the possibility to perform the procedure after several hours, the low radiation dose for patients and operators, and the small amount of time that is required to perform it.

## Figures and Tables

**Table 1 diagnostics-13-02628-t001:** A summary of the cited studies that used ^99m^Tc-albumin macroaggregates for lung radio-guided surgery.

Activity (MBq)	Type of Lesion	Mean Size of Lesion (mm)	Histology of Lesions	Average Length of Surgery (minutes)	Reference
37–111	32 solid nodules (2 subsolid, 2 non-solid, and 2 wall/pleural)	10.1 ± 4.9	6 benign lesions, 8 primary lung cancers, and 24 metastases	19	Vollmer et al. [24]
7–10	2 GGOs and 3 part-solid nodules	7	10 benign lesions (neoplastic and non-neoplastic), 4 malignant lung neoplasms, and 10 lung metastases	102.5	Carvajal et al. [25]
11.1	29 nodules (12 GGOs)	8	4 benign lesions, 15 primary lung cancers, and 10 metastases	85.5	Dailey et al. [26]
15	15 GGOs, 12 solid nodules, and 10 partly solid nodules	11	15 primary lung cancers, 8 metastases, and 13 benign lesions	-	Durmo et al. [27]
2–9	SPNs/GGOs with a distance from the nearest pleural surface of 20–40 mm	9 ± 4	8 primary lung cancers, 4 metastases, and 7 benign lesions	-	Bertolaccini et al. [28]
5–10	SPNs or multiple small pulmonary nodules at least 5 mm from the visceral pleura	8.3	11 primary lung cancers, 7 metastases, and 21 benign lesions	50	Boni et al. [29]
5–10	SPNs distant from the nearest pleura between 1.5 and 3 cm	11	23 primary lung cancers, 9 metastases, and 18 benign lesions	41 for group A and 43 for group B	Gonfiotti et al. [30]
7–15	SPNs with a distance from visceral pleura of 1 cm	9	14 benign lesions, 85 primary lung cancers, and 24 metastases	-	Galetta et al. [11]
5	SPNs deeper than 5 mm below the visceral pleura	13	206 benign lesions, 130 metastases, and 59 lung primary tumors	40	Manca et al. [31]
2–9	Posterior location and non-solid or partly solid nodules	9 ± 4	7 benign lesions, 4 metastases, and 8 lung primary tumors	6 min for radio localization	Bertolaccini et al. [33].
5–10	GGOs or SPNs with a distance from visceral pleura more than or equal to 5 mm and within 3 cm	13	218 benign lesions, 149 metastases, and 84 lung primary tumors	40 (3 for radio localization)	Ricciardi et al. [34]
5–10	SPNs with a distance from the nearest pleural surface more than or equal to 10 mm	13	15 benign lesions, 55 primary lung cancers, and 105 metastases	44	Davini et al. [8]

RALN: radiotracer-assisted localization of lung nodules; SPN: solitary pulmonary node.

## Data Availability

Not applicable.

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
