# Peer review of "Radio-Guided Lung Surgery: A Feasible Approach for a Cancer Precision Medicine"

_diagnostics, 2023, doi:10.3390/diagnostics13162628_

Round 1

Reviewer 1 Report

Thank you very much for selecting me as a reviewer for this article, entitled " Radio-guided lung surgery: a feasible approach for a cancer precision medicine".

I congratulate the authors good written paper.

This review is interesting and described radio-guided surgery as an approach for localizing ground glass opacities, lung nodules, ad metastatic lymph nodes. Small lung lesions are often a challenge for a surgeon.

Main issues:

1. English language should be edited, for example line 64 mammalian

2. Line 17 : AND - why capital letters ?

3. Line 119  The mean lenght of the surgery was 19 minutes ? VATS or thoracotomy in 19 minutes ?

Best regards

Author Response

I would thank the reviewer for the precious observations. We have been glad to put them in place, conscious of the fact that your advice and suggestions will certainly improve the quality and comprehensibility of our work. All recommended corrections have been made and underlined in the manuscript, as detailed below:

  1. The English language has been edited throughout the text. In particular, the sentence in line 64 has been modified as follows: “Other GGO lesions with low glucose uptake are lung metastasis from breast cancer or renal carcinoma but also mucinous adenocarcinomas of the gastrointestinal tract that are characterized by mucinous components and low cellularity”.
  2. Line 17: AND was written in lowercase letters.
  3. Line 119: the mean time of 19 minutes was referred to VATS. The sentence has been corrected in “The mean length of VATS was 19 minutes”.

Moreover, references 14, 23, 25, 28, and 46 have been corrected.

Reviewer 2 Report

Thank you for the opportunity to analyze your article.       

In this article, authors have reviewed different technics of “radio-guided surgery” which a burden topic today. As they have highlighted, smaller lesions, non-palpable lesions and minimally invasive surgical approaches, handless approaches required imaging assistance to perform a targeted and tailored-made resection with the best oncological outcomes and sparing lung function to preserve quality of life.  As a paradox, “resecting less”, smaller lesion, non-palpable lesion as GGO, with “small incisions” required more, more imaging and more surgical devices. 

            Concerning the review of introduction:

            The introduction is well written, with a good synthesis about early non-small cell lung cancer diagnosed with screening programs. 

Line 80 to 83: more than the VATS approach, this is the concept of minimally invasive surgery, starting with VATS and dealing with robotic also today which “changed the management of these lesions” as you have mentioned. 

Lines 86 to 88: International guidelines from different oncological lung societies have define the lung surgical oncological resection as an anatomical lung resection associated to a complete lymph node dissection. Selective, Oriented lymph node dissection, sampling lymph node assessment are still discussed, this is a very burden subject but no recommendations’ modifications have been made today. Please modify this paragraph. 

Line 96: “in order to enhance post operative courses”, to preserve the best lung function, but after having properly diagnosed and “cured” the lung cancer. “Thoracotomy is still not a crime!”

            Concerning the methodology of the review: 

            No major concerns about it. 

            Concerning the review of 99mTc-albumin macroaggregates in radio-guided surgery of lung lesions:

            An exhaustive review of major surgical series have been made with retrospective studies and studies comparing 99mTc-albumin to, hook wire, or dye injection. Main results are well reported with complications, and surgical success rate and post operative histology and failed explications. Professional exposure may be more highlighted cause it’s a major concerns for many procedures and many year of practice. 

            Maybe a synthetic results table could be very interesting and illustrative with the main results. 

Concerning the review of Radio-guided sentinel lymph nodes biopsy:

            Less studies have been reported and for lymph node assessement, ICG is a new topic of interest compared to radio-guided approach. 

            No other studies have been found? 

            Concerning the review of Cyanoacrylate combined to Technetium-99m-sulfur colloid (Sn-99mTc): 

            This approach presents the advantage to prevent radiotracer dispersion. No other interesting reports have been found? 

Concerning the review of Indium-111 (111In)-pentetreotide for neuroendocrine tumors:

No major concerns about this technic which is not often performed.

Concerning the review of Fluorine-18 (18F)-deoxyglucose:

As you have mentioned the challenge is to distinguish FDG activity between the lesion and the lymph nodes for LN assessment. 

Concerning the review of Naphthalocyanine and Cupper-64 (64Cu) :

             An interesting technic, no human applications have been reported? 

            Concerning the discussion and the conclusion:

Your discussion looks like your conclusion. 

You have reported a good review dealing with radio-guided surgery, and maybe a table with main results of the most important series could be very interesting and informative.

But you only present these technics without mentioning other technic, just briefly as dye marking and ICG marking thru a bronchoscopy approach for example. Because, why performing “radio-guided surgery? Compared to “lung-endoscopic-guided surgery” for example? This approach is center, multidisciplinary team and expertise dependent. And the choice of the guided-approach is also impacted by the availability of the technical radiological unit or the operating room. For example in my hospital, radio-guided surgery can be performed, with my colleagues from the unit of radiology. The patient is admitted the day before the surgery or the day of the surgery, and due to the “volume pressure” in the operating room, everything that can be made out of the OR is good for the all organization. But in my precedent hospital, this can’t be made, and we used to perform a “bronchoscopic-guided surgery”.

Need to highlighted the idea of team working with a shared technical devices, under the pressure of the bed occupancy and capacity. 

            It’s a well written, easy reading and interesting review, that just need some illustrative precisions.

            Congratulations to authors for this work. 

Author Response

I want to thank the reviewer for the attention shown and the precious suggestions made, which we have welcomed. We have been glad to put them in place, conscious of the fact that your advice and suggestions will certainly improve the quality and comprehensibility of our work.

The recommended corrections have been underlined in the manuscript, as detailed below:

  1. Lines 80-83, the sentences have been reformulated as follows: “The advent of Video-assisted thoracoscopic surgery (VATS) and robotic approaches changed the management of these lesions since it represents the first choice for peripheral lung nodules, even if its application becomes challenging in the case of small, nonsolid, or deep lung nodules”;
  2. Lines 86-88, the paragraph has been modified: “It is well-known the benefit with respect to the identification of sentinel lymph nodes (SLN) in patients with different solid tumors, such as breast cancer and melanoma in which the SNL dissection predicts the status of the more distant lymph nodes stations. On the other hand, the application of this technique is still debated in lung cancer. In fact, international recommendations suggest that a complete lymph node resection has been done and no changes have been made so far”;
  3. Line 96, the sentence has been modified as follows: “Intraoperative localization of these entities through the support of radiopharmaceutical agents represents a reliable and easy technique to identify lung lesions and obtain a satisfactory surgical resection, and in order to enhance post-operative courses”;
  4. Professional exposure has been highlighted in the following lines: lines 147-151: “A notable aspect is the radiation exposure of involved personnel… relatively lower with radiotracer-assisted surgery”; lines 160-161: “Moreover, [99mTc]-MAA has a non-significant radiation exposure for patients and hospital personnel.”; lines 171-173: “As stated before, the low patient-absorbed radiation dose of the technetium, thanks to its 6 hours of half-life, makes this tracer a safe choice for radioguided surgery”; lines 191-192: “The minimal used activity did not require any special exposure reading for the staff.”; lines 243-244: “Finally, as observed in the previous studies, the low dose of the tracer made this procedure safe for the involved personnel.”; lines 252-253 “Also in this experience, the choice of 99mTc with its low absorbed radiation made this procedure safe for the patient and for the involved staff.”.
  5. A table has been added to summarize the paragraph “99mTc-albumin macroaggregates in radio-guided surgery of lung lesions”.
  6. Concerning the ICG approach and studies on Cyanoacrylate combined with Technetium-99m-sulfur colloid, even if this aspect could be surely interesting to better analyze, no other studies have been found with the chosen keywords “radioguided surgery and lung cancer”.
  7. Regarding Naphthalocyanine radiolabeled with Cupper-64, no studies reporting application on humans have been found.
  8. The discussion has been ameliorated and a comparison with ICG and ultrasonography has been mentioned.

Moreover, references 14, 23, 25, 28, and 46 have been corrected.